# An Improved Chromosome-Level Genome Assembly of the Firefly *Pyrocoelia pectoralis*

**DOI:** 10.3390/insects15010043

**Published:** 2024-01-08

**Authors:** Xinhua Fu, Victor Benno Meyer-Rochow, Lesley Ballantyne, Xinlei Zhu

**Affiliations:** 1College of Plant Science and Technology, Huazhong Agricultural University, Wuhan 430070, China; 2Firefly Conservation Research Centre, Wuhan 430070, China; zhuxinlei32@gmail.com; 3Department of Ecology and Genetics, Oulu University, SF-90140 Oulu, Finland; meyrow@gmail.com; 4Agricultural Science and Technology Research Institute, Andong National University, Andong 36729, Republic of Korea; 5School of Agricultural, Environmental and Veterinary Sciences, Charles Sturt University, P.O. Box 588, Wagga Wagga 2678, Australia; lballantyne@csu.edu.au

**Keywords:** firefly, chromosome-level genome assembly, courtship behavior, evolution, conservation, *Pyrocoelia pectoralis*

## Abstract

**Simple Summary:**

The nocturnal firefly *Pyrocoelia pectoralis* is an endemic and endangered species in China. To more fully understand the role of sexual dimorphism and the evolution of courtship behavior in this species and improve its conservation, an improved chromosome-level genome assembly of *P. pectoralis* was conducted, and a high-quality draft of the genome was generated. Our research provides an insight into the evolution of courtship behavior in fireflies.

**Abstract:**

The endemic and endangered Chinese firefly *Pyrocoelia pectoralis* is a sexually dimorphic, nocturnal species. A previous attempt by this team to assemble a draft genome of *P. pectoralis* using PacBio and Illumina HiSeq X Ten platforms was limited in its usefulness by high redundancy and contamination. This prompted us to conduct an improved chromosome-level genome assembly of *P. pectoralis*. Ten chromosomes were further assembled based on Hi-C data to a 532.25 Mb final size with a 52.87 Mb scaffold N50. The total repeat lengths in the genome of *P. pectoralis* amount to 227.69 Mb; 42.78%. In total, 12,789 genes could be functionally annotated using at least one public database. Phylogenetic inference indicated that *P. pectoralis* and *P. pyralis* diverged ~51.41 million years ago. Gene family expansion and contraction analysis of 12 species were performed, and 546 expanded and 2660 contracted gene families were identified in *P. pectoralis*. We generated a high-quality draft of the *P. pectoralis* genome. This genome assembly should help promote research on the species’ sexual dimorphism and its unique courtship behavior, which involves a combination of pheromonal and bioluminescent signals. It also can serve as a resource for accelerating genome-assisted improvements in the conservation of this species.

## 1. Introduction

Fireflies (Coleoptera: Lampyridae) are the most common representatives of terrestrial bioluminescent animals [1,2,3]. The firefly *Pyrocoelia pectoralis* (Olivier) (Lampyridae: Lampyrinae) is endemic to China and has terrestrial larvae, which play an important ecological role as biological agents to control the land snail *Bradybaena ravida* [4]. The snail is widely distributed throughout China, Japan, Korea, and Russia, and the species is known to cause considerable damage to various vegetables, peaches, grapes, and corn [5]. During courtship, the stationary females of *P. pectoralis* glow continually and are presumed to also emit a sex pheromone to attract flying, bioluminescent males [6]. The adults of this species are sexually dimorphic (Figure 1) with wingless females that are considerably larger than the males (♂ 15 mm versus ♀ 25 mm total body lengths). In many ways, their pre-copulation behavior resembles the firefly *Rhagophthalmus ohbai* Wittmer, which was analyzed and described by Lau et al. [7], who showed that the males in search of their wingless females use their large eyes with elevated spectral sensitivity to light of longer wavelengths in the ventral eye half to detect the yellowish light emitted by the females on the ground.

Fireflies generally and the species *P. pectoralis* in particular are highly appreciated for their luminescence and in some places, the bioluminescence of these insects attracts tourists trying to see and photograph the spectacle. To lose the species would, therefore, be a double tragedy. The second reason why this species ought to be saved from becoming extinct is a non-applied purely zoological one related to basic science, as the phylogenetic relationships between lampyrid species and their evolutionary histories have still not been fully worked out. For the two reasons explained above, a study focusing on this firefly’s genome is, therefore, timely and important.

As the cost of sequencing continuously decreases, genomic-scale data generation through high-throughput sequencing technologies can now routinely be used in studies on reproduction, phylogenetic relationships, and species delimitation [8]. High-throughput sequencing technologies in genomics can enhance insect breeding and support diversity studies and conservation goals by generating suitable genetic markers [9,10,11]. To explore the evolution of courtship signals in fireflies, genome evolutionary analysis, such as the evolution of genome size and the expansion and contraction of gene content in *P. pectoralis*, becomes necessary. Using PacBio and Illumina HiSeq X Ten platforms, the first firefly draft genome (original genome Ppec-1.0) was assembled (genome size 760.4 Mb, scaffold N50 3.04 Mb) (Appendix A) [12]. We detected non-firefly genes such as mitochondrial, bacterial, viral, fungal, and other no-hit genes (Appendix A) when we performed annotation of the original genome Ppec-1.0. Additionally, BUSCO assessment showed the completeness of complete and single-copy in the original genome, Ppec-1.0 is low at 60.8%, while complete and duplicated BUSCOs (D) are high at 38.3% (Appendix A). Thus far, contamination and high redundancy have limited its usefulness.

In this study, we used Hi-C technology to improve the *P. pectoralis* genome assembly (Hi-C genome Ppec-2.0) in the chromosomal level. We also assembled and annotated its mitochondrial genome. Gene family evolution and phylogenetic reconstruction analyses were performed. Collectively, our findings represent a valuable resource for studies on the species’ sexual dimorphism and its unique courtship behavior that uses a combination of pheromonal and bioluminescent signals. They can also serve as a resource for accelerating genome-assisted improvements in conservation.

## 2. Materials and Methods

### 2.1. Sample Collection and Feeding Scheme

Male and female individuals of *P. pectoralis* were lab-reared for two generations at the Huazhong Agricultural University in Wuhan (China). The original population was collected in Ezhou City, Hubei Province, in October 2018. Larvae were bred in transparent plastic boxes (20 cm diameter × 6 cm high) and provided with crushed land snails (*B. ravida*) as prey [7].

### 2.2. Karyotype Analysis

Mitotic and meiotic chromosomes were obtained from the gonads of ten fifth instar larvae following [13]. The gonads were removed into a 10 mg/mL colchicine solution (in insect saline solution) for 120 min and then subjected to hypotonic treatment for 30 min. All the gonads were fixed in Carnoy I (three parts methanol and one part acetic acid), for 60 min. For preparation of the slides, the gonads were macerated in 45% acetic acid until a cell suspension was acquired, which was then spread over a slide and dried on a metal plate at 40 °C. To determine the number, size, and morphology of chromosomes, the slides were stained with DAPI for 10 min. Images were obtained with a Leica TCS SP8 confocal microscopy station (Leica Camera, Wetzlar, Germany). Adobe Photoshop (version 2021, Adobe, San Jose, CA, USA) was used to arrange karyotypes, and karyotypes were organized in decreasing order of size. For the karyotype analysis, the chromosomes of *P. pectoralis* were stained with DAPI (blue).

### 2.3. Hi-C Library Preparation and Chromosome Assembly by Hi-C Data

We previously applied the Illumina HiSeq X Ten and PacBio platforms to sequence the genome of *P. pectoralis* [12]. However, high redundancy and contamination were detected and limited its usage. To reduce sequence redundancy and contamination, we performed a chromosome-level assembly. We constructed the Hi-C library and obtained sequencing data via the Illumina Novaseq platform. A whole fresh adult female *P. pectoralis* body was vacuum-infiltrated in a nuclei-isolation buffer supplemented with 2% formaldehyde. The fixed tissue was then ground to powder before re-suspending in a nuclei-isolation buffer to obtain a suspension of nuclei. The purified nuclei were digested with 100 units of DpnII and marked by incubating with biotin-14-dCTP. The ligated DNA was sheared into 300–600 bp fragments and was then blunt-end repaired and A-tailed, followed by purification through biotin–streptavidin-mediated pulldown. Finally, the Hi-C libraries were quantified and sequenced using the Illumina Novaseq platform.

In total, 594 million paired-end reads were generated from the libraries. Quality controlling of Hi-C raw data was performed using Hi-C-Pro [14]. Firstly, low-quality sequences (quality scores < 20), adaptor sequences, and sequences shorter than 30 bp were filtered out using fastp, and then the clean paired-end reads were mapped to the draft assembled sequence using bowtie2 (v2.3.2) (-end-to-end --very-sensitive -L 30) to obtain the unique mapped paired-end reads. Valid interaction paired reads were identified and retained by HiC-Pro (v2.8.1) [14] from unique mapped paired-end reads for further analysis. Invalid read pairs, including dangling-end, self-cycle, re-ligation, and dumped products were filtered by HiC-Pro (v2.8.1). The scaffolds were further clustered, ordered, and oriented onto 10 pseudo chromosomes by LACHESIS (https://github.com/shendurelab/LACHESIS (accessed on 18 January 2022)), with parameters CLUSTER_MIN_RE_SITES = 100, CLUSTER_MAX_LINK_DENSITY = 2.5, CLUSTER NONINFORMATIVE RATIO = 1.4, ORDER MIN N RES IN TRUNK = 60, and ORDER MIN N RES IN SHREDS = 60. Finally, placement and orientation errors exhibiting obvious discrete chromatin interaction patterns were manually adjusted. These 10 pseudochromosomes correspond to the 10 chromosomes for *P. pectoralis*. To check the completeness and quality of the assembly, BUSCO version 5.1.3 [15,16] was used to search the 1367 benchmarking universal single-copy orthologous genes in insecta_odb10.

The mitochondrial genome (mtDNA) was assembled separately from the nuclear genome using Illumina HiSeq X Ten short reads [12]. The clean reads were used to produce a de novo assembly using IDBA-UD [17], with minimum and maximum k values of 41 and 141 bp, respectively. The mitogenome sequence of *P. pectoralis* was identified by Gneious 10.1.3 (http://www.geneious.com, accessed on 25 May 2022). Genomic annotations were performed using MITOZ v2.3 [18] and tRNAscan-SE 2.0 [19]. Tandem repeats were identified using the Tandem Repeat Finder v. 4.09 [20]. A map of the complete mitochondrial genome was generated using Proksee (CGView) (https://proksee.ca/, accessed on 8 November 2023) [21].

### 2.4. Genome Annotation

The simple repeat sequences (SSRs) of the Hi-C genome Ppec-2.0 were analyzed using GMATA v2.2 [22] software, while a Tandem Repeats Finder (TRF v4.07b) [20] recognized all tandem repeat elements in the Hi-C genome Ppec-2.0. To obtain a better estimation of tandem repeats, the putative satDNA elements of the IlluminaHiSeq X paired-end unassembled short read sequences were identified using the RepeatExplorer and TAREAN pipelines in the Galaxy platform [23].

Transposable elements (TEs) in the *P. pectoralis* genome were then identified using a combination of ab initio and homology-based methods. Briefly, an ab initio repeat library for *P. pectoralis* was first predicted using MITE-hunter [24] and RepeatModeler (v1.0.11; http://www.repeatmasker.org/RepeatModeler/, accessed on 20 April 2022) with default parameters, in which LTR_FINDER [25], LTR_harverst [26], and LTR_ retriever [27] were used to obtain as much reliable information as possible on long terminal repeat (LTR) retrotransposons. The obtained library was then aligned to the TE class Repbase (http://www.girinst.org/repbase, accessed on 20 April 2022) to classify the type of each repeat family. For further identification of the repeats throughout the genome, RepeatMasker version 4.0.7. (http://www.repeatmasker.org, accessed on 20 April 2022) was applied to search for known and novel TEs by mapping sequences against the de novo repeat library and Repbase TE library. Overlapping transposable elements belonging to the same repeat class were collated and combined.

Three independent approaches, including ab initio prediction by AUGUSTUS v3.3.1 [28], a homology search by GeMoMa v1.6.1 [29], and reference-guided transcriptome assembly by software PASA v2.3.3 [30] were used for gene prediction in a repeat-masked genome. Gene function information, motifs, and domains of their proteins were assigned by comparing them with public databases including SwissProt, NR, KEGG, KOG, and Gene Ontology. Transfer RNAs (tRNAs) were predicted using tRNAscan-SE v2.0 with eukaryote parameters. MicroRNA, rRNA, small nuclear RNA, and small nucleolar RNA were detected using Infernal v1.1.2 cmscan [31] to search the Rfam database [32]. The rRNAs and their subunits were predicted using RNAmmer v1.2 [33].

### 2.5. Identification of Homologous and Orthologous Gene Sets

To identify homologous relationships among *P. pectoralis* and 11 other insects, information available on *Drosophila melanogaster* (GCA_000001215.4), *Onthophagus taurus* (GCA_000648695.2), *Leptinotarsa decemlineata* (GCA_000500325.2), *Anoplophora glabripennis* (GCA_000390285.2), *Tribolium castaneum* (GCA_000002335.3), *Agrilus planipennis* (GCA_000699045.2), *Dendroctonus ponderosae*(GCA_000346045.2), *Ignelater luminosus* (GCA_011009095.1), *Abscondita terminalis* (GCA_013368085.1), *Lamprigera yunnana* (GCA_013368075.1), and *Photinus pyralis* (GCA_008802855.1) was downloaded from NCBI (https://www.ncbi.nlm.nih.gov/genome, accessed on 30 August 2021). The gene sets were then aligned using OrthMCL v2.0.9 [34]. Protein sets were collected from 12 sequenced insect species, and the longest transcripts of each gene were extracted, in which miscoded genes and genes exhibiting premature termination were discarded. The extracted protein sequences were then aligned pairwise to identify conserved orthologs using Blastp v2.6.0+ set to an E-value threshold of ≤1 × 10^−5^, and orthologous inter-genome gene pairs, paralogous intra-genome gene pairs, and single-copy gene pairs were further identified using OrthMCL.

### 2.6. Phylogenetic Analysis

On the basis of the identified orthologous gene sets with OrthMCL v2.0.9 [34], a molecular phylogenetic analysis was performed using the shared single-copy genes. Briefly, the coding sequences were extracted from the single-copy families, and each ortholog group was multiple-aligned using Mafft v7.313 [35]. Poorly aligned sequences were then eliminated using Gblocks v0.91b [36], and the GTRGAMMA substitution model of RAxML v8.2.10 [37] was used for the phylogenetic tree construction with 1000 bootstrap replicates. The generated tree file was displayed with MEGA CC v10.1.8 [38]. Based on the phylogenetic tree, the RelTime of MEGA-CC was utilized to compute the mean substitution rates along each branch and estimate the species’ divergent time. Three fossil calibration times were obtained from the Time Tree database (http://www.timetree.org/, accessed on 20 April 2022) as the time control, including the divergence times of *Drosophila melanogaster* versus *Tribolium castaneum*, in which the estimated time is 308 MYA (234–370 MYA), and *Ignelater luminosus* versus *Photinus pyralis*, in which the estimated time is 133 MYA (103–127 MYA).

### 2.7. Species-Specific Genes and Gene Family Expansion and Contraction

Proteins with no homologs in the other 11 insect genomes were extracted as species-specific genes, including *P. pectoralis*-specific unique genes and unclustered genes. Functional annotation of species-specific genes and enrichment tests were performed using information from homologs in the Gene Ontology (http://www.geneontology.org/, accessed on 18 January 2022) and KEGG (Kyoto Encyclopedia of Genes and Genomes) databases.

TE expansion and contraction of gene families were determined using CAFE v5.0.02937 [39]. The results from the phylogenetic tree with divergence times were used as inputs. A *p*-value of 0.05 was used to identify families that were significantly expanded and contracted. Gene ontology (GO) enrichment of expanded orthogroups and species-specific orthogroups of *P. pectoralis* were analyzed and visualized by REVIGO v1.8.1 [40].

## 3. Results

### 3.1. Chromosome-Level Genome Assembly

We obtained high-quality metaphase chromosome spreads with large metaphase areas, large numbers of chromosomes, and few chromosome overlaps (Appendix A). Ten pairs of chromosomes were observed (2n = 20) (Appendix A) and showed that this individual was most probably a female. We quantified our observations and found that 52.3% of 21 mitotic plate cells had 19 or 20 isolated chromosomes. However, there was no evidence of a Y chromosome. We used Hi-C technology to improve the genome assembly to the chromosomal level. A total of 83.42 Gb of high-quality sequencing data was generated from a 350 bp insert size Hi-C library, and the quality assessment of the Hi-C data is shown in Appendix A. In total, removing redundant genes and contamination led to a final genome assembly of 532.25 Mb, containing 363 scaffolds with N50 of 52.87 Mb (Appendix A). The BUSCO assessment indicated that the completeness of the Hi-C genome Ppec-2.0 is 94.7%, lower than the original genome Ppec-1.0 with 99.1% (Appendix A). We super-scaffolded the *P. pectoralis* genome assembly into 10 pseudo-chromosomal linkage groups (Appendix A), with a size of 521.95 Mb and 337 scaffolds (98.07%) anchored on all chromosomes (Figure 2a, Appendix A).

### 3.2. Mitochondrion Assembly

The complete mitogenome of *P. pectoralis* is a typical circular molecule with a total length of 16,792 bp (Appendix A). Like most Coleopteran insects, there are thirty-seven coding genes, including thirteen protein coding genes (PCGs), two rRNAs, twenty-two tRNAs, and one major control region. Among thirty-seven genes, *nad5*, *nad4*, *nad4l*, and *nad1*, eight tRNAs (tRNA-Gln, Cys, Tyr, Phe, His, Pro, Leu (CUN), Val) and two RNAs were located on the reverse strand, *nad2*, *cox1*, *cox2*, cox3, *atp6*, *atp8*, *nad3*, *nad6*, and *cytb*, and fourteen tRNAs were located on the direct strand (Appendix A).

The maximum proportion of the mitogenome is taken up by PCGs, occupying 66.08% of the whole mitogenome. The rRNA and tRNA accounted for 11.95% and 8.42%, respectively, and the non-coding regions accounted for 13.54%. There are two A+T-rich regions in the mitogenome, one having six tandem repeats with a period size of 133 bp, while the other is the control region containing the origin of replication.

From the codon usage analysis, seven PCGs, including *atp6*, *cox2*, *cox3*, *cytb*, *nad3*, *nad4*, and *nad4l*, used the same ATG/ATA as the starting codon. In the other six PCGs, starting codons are different, i.e., *nad1* used TTG, *nad6* used ATC, and ATT was used in four other PCGs. Only five PCGs used the complete termination codon, such as TAA or TAG; the other PCGs used the incomplete termination codon, such as T.

### 3.3. Genome Annotation

Repeated sequences were mined and annotated. In *P. pectoralis*, the total length of the repeats is 227.69 Mb (42.78% of the genome) (Appendix A). With a value of 42.78%, the total repeats ratio of *P. pectoralis* occupies the middle ground but is lower than the ratios of *Lamprigera yunnana* (65.37%) and *Photinus pyralis* (46.33%) and higher than *A. terminalis* (33.76%) and *Aquatica lateralis* (27.46%) [41,42] (Appendix A). To be specific, the number of TEs (transposable elements) was identified to be 839,559, with the sequence percentage being 38.8%. Among these TEs, the dominant repetitive sequence type involves the DNA transposon, accounting for 18.21% (~96.92 Mb). The detailed statistics of TEs are listed in Appendix A. Four major types of TEs are identified and compared with other firefly species (Appendix A). Sequence divergence rate analyses showed that TE sequences of *P. pectoralis* form a peak with a low divergence rate of ~2.3% (Appendix A), indicating a recent expansion of TEs in *P. pectoralis*.

The content of tandem repeat elements was analyzed and compared between Illumina HiSeq X paired-end unassembled short read sequences and Hi-C genome Ppec-2.0. Appendix A shows the size distribution of superclusters and information about the number of reads that were actually analyzed in unassembled Illumina sequences. From the 1,360,133 analyzed reads, 23,488 reads in clusters were annotated as organellar (mitochondrial or plastid) sequences from sequencing control DNA, leaving 1,336,645 reads representing nuclear DNA (Appendix A). Considering the read length of 120 bp, the analyzed reads represented 0.307× coverage of the nuclear genome, providing sufficient sensitivity to analyze highly and moderately repeated sequences. The analysis revealed that satellite DNA accounts for up to 1.56% of the genome (20,857 reads) (Appendix A). Meanwhile, a total of 16,467 (0.04% of the genome) simple repeat sequences (SSRs) was identified in the Hi-C genome Ppec-2.0, as well as a total of 20,713 (0.88% of the genome) tandem repeat sequences (Appendix A).

The total of the protein-coding genes is much lower than the original genome Ppec-1.0 with 23,092 protein-coding genes [12]. We defined the models of protein-coding genes in the Hi-C genome Ppec-2.0 using the de novo prediction, transcriptome, and homology-based methods, producing a total of 13,292 protein-coding genes with an average gene length of 14,806 bp and CDS 1481 bp (Appendix A). The average exon number per gene was 5.18. The average intron length was 3188 bp (Appendix A). The total number of coding genes was smaller than in *L. yunnana*, *A. terminalis*, and *P. pyralis*, where the respective numbers were 19,443, 20,436, and 20,646; however, the number is close to that of other insects. This may be related to the degree of redundancy during genome assembly. The values of other parameters were close to those of the published genomes (*P. pyralis*, *L. yunnana*, and *A. terminalis*), indicating the reliability of the annotation results.

The genes for which Swissprot annotations were obtained were 10,231, and the number of genes for which KEGG annotation was obtained was 6491. GO annotation obtained 8002, NR annotation obtained 12,369, and KOG annotation obtained 8745. In summary, 12,789 genes (96.22%) could be functionally annotated using at least one public database (Swissprot, NR, KEGG, GO, and KOG) (Appendix A). The completeness of proteins for the predicted genes in the Hi-C genome Ppec-2.0 was evaluated by BUSCO with Insecta_odb10 (n = 1367), and there are 1241 complete BUSCOs (90.8%), while the complete BUSCOs of the original genome Ppec-1.0-predicted proteins are 1295 (95.2%) (Appendix A). Four types of ncRNAs, including 434 rRNAs, 533 microRNAs (miRNAs), 190 cis-regulatory elements, and 2898 transfer RNAs (tRNAs), were also identified (Appendix A).

### 3.4. Gene Family Identification and Phylogenetic Relationships

To infer the evolutionary status and trace the phylogenetic placement of *P. pectoralis*, gene family clustering was performed using OrthoMCL. The gene family clusters were divided into five categories: (1) multiple-copy orthologs have multiple copies in one species; (2) single-copy orthologs have only one copy in one species; (3) the other orthologs are the rest of the orthologs; (4) unclustered genes have no homology with others; and (5) unique paralogs are genes that only exist in one specific species (Appendix A). A total of 9543 gene families (12,325 total genes) was identified. *P. pectoralis* contained 3016 multiple-copy ortholog genes, 268 unique genes, and 967 unclustered genes. There are 1235 species-specific ortholog genes in *P. pectoralis*. In addition, 1709 single-copy ortholog genes were defined in all species, which were aligned to develop a super-sequence for each species that was used to construct a phylogenetic tree (Appendix A). Phylogenetic results showed that *P. pectoralis* is closely related to *P. pyralis* of the subfamily Lampyrinae. Phylogenetic inference also indicated that *L. yunnana* represents a sister taxon to Luciolinae and that *P. pectoralis* and *P. pyralis* diverged ~51.41 million years ago (Figure 2b).

### 3.5. Patterns of Gene Family Expansion and Contraction

Our analysis of gene family expansion and contraction of the 12 species identified 546 expanded and 2660 contracted gene families (*p*-value < 0.05) in *P. pectoralis* (Figure 2b). GO and KEGG enrichment analyses revealed that in the *P. pectoralis*-expanded gene, families were enriched by the nucleotide-binding pathway (including DNA replication, DNA recombination, nucleosome assembly, DNA-templated transcription, initiation, and DNA integration in the nucleosome), the oxidoreductase pathway (including oxidoreductase activity, oxidoreductase activity, acting on the CH-CH group of donors, 2-alkenal reductase [NAD(P)+] activity, and 15-oxoprostaglandin 13-oxidase activity), and some important receptors related to signal transduction pathways (including ionotropic glutamate receptor activity, olfactory receptor activity and sensory perception of smell) (Appendix A). Conversely, *P. pectoralis* showed significantly contracted gene families in the ABC transporters process, bile secretion process, cAMP signaling pathway, and fatty acid biosynthesis process (Appendix A). In addition, odorant binding genes (GO: 0005549) were contracted.

The GO and KEGG enrichments in *P. pectoralis* also showed that species-specific orthologous genes were enriched in proteolysis, odorant binding, oxidoreductase and nucleic acid-related genes, which were contained in connection with the xenobiotic metabolism (*p*-value = 4.63 × 10^−9^), steroid hormone biosynthesis (*p*-value = 7.48 × 10^−7^), insect hormone biosynthesis (*p*-value =1.29 × 10^−6^), retinol metabolism (*p*-value = 7.88 × 10^−6^), and the neuroactive ligand–receptor interaction pathway (*p*-value = 1.22 × 10^−4^) (Appendix A). GO enrichment analysis revealed that the *P. pectoralis*-expanded gene families, and species-specific orthologous genes are both enriched in the same molecular functions, such as odorant binding and oxidoreductase activity (Figure 3).

## 4. Discussion

We reported a high-quality chromosome-level genome assembly of *P. pectoralis*, with a final assembly length of 532.25 Mb. Compared with other firefly genomes, our genome assembly exhibited the least total scaffold number (363) and the highest scaffold N50 (52.87 Mb), while other firefly genome total scaffold numbers ranged from 2160 to 5365, and scaffold N50 in other firefly genomes varied from 0.69 Mb to 47.02 Mb [43]. The original genome Ppec-1.0 has considerably more protein-coding genes than the Hi-C genome Ppec-2.0. Meanwhile, the BUSCO assessment of the original genome Ppec-1.0 is also higher than the Hi-C genome Ppec-2.0. However, the BUSCOs of the complete and single-copy of the original genome Ppec-1.0 are much lower at 60.8% than 92.1% of the Hi-C genome Ppec-2.0. The complete and duplicated BUSCOs (D) of the original genome Ppec-1.0 are much higher than the Hi-C genome Ppec-2.0. Thus, the results revealed that deduplication in the Hi-C genome Ppec-2.0. has been successful. This demonstrates that we were able to successfully reveal the high quality of this genome in terms of continuity and completeness, and this is a valuable asset in connection with an in-depth study of the courtship behavior of the firefly, as well as the conservation efforts involving this firefly species.

The genome size of *P. pectoralis* is larger than *A. terminalis* and *P. pyralis* but smaller than *L. yunnana* and *Aquatica lateralis*, although only *P. pectoralis* and *Photinus pyralis* were assembled regarding chromosome level [41,42]. The result indicates that genome size variations arise mainly from the relative abundance of TEs, especially DNA and LINEs, which are also the two most abundant types of TEs previously reported from the genomes of all luminous beetles [41,42].

Although we obtained DAPI images from different individuals representing different stages of mitotic and meiotic chromosomes (Appendix A), the Y chromosome was not evident, and the karyotype was still not clear. It is planned that fluorescence in situ hybridization (FISH) will be carried out to determine the karyotype of *P. pectoralis* and to predict which linkage group sequences (LG) occur in each chromosome.

Most fireflies use light flashes or a continuous glow during courtship [1,2,3,6,9]. Pheromonal signals, in addition to weak or even absent bioluminescence signals, exist in several diurnal species [2], including *Lucidina biplagiata*, *Lucidota atra*, *Pyropyga nigricans*, *Photinus indictus*, and *Phosphaenus hemipterus*. A comparison between nearly identically sized species of luminescent *Aquatica lateralis* (as *Luciola lateralis*) and the non-luminescent *Lucidina biplagiata* by Meyer-Rochow [44] showed that the males of the luminescent species had significantly larger eyes than those of the non-luminescent species. In addition to bioluminescence, females of several lampyrid taxa attract males using a combination of pheromonal and bioluminescent signals [2], e.g., *Pyrocoelia rufa*, *P. pectoralis*, *Pleotomus pallens*, and *Phausis* spp.

Gene family expansion identified 546 expanded gene families in *P. pectoralis*. GO and KEGG enrichment analyses revealed that the *P. pectoralis*-expanded gene families were enriched by the nucleotide-binding pathway, the oxidoreductase pathway, and some important receptors related to signal transduction pathways. On this basis, we suggest that *P. pectoralis* has novel genes through gene family expansion and species-specific orthologous genes to evolve a unique courtship behavior that uses a combination of pheromonal and bioluminescent signals. Thus, our research provides an insight into the evolution of courtship behavior in fireflies, and being aware of these details should enable conservationists to more easily identify habitat characteristics that would facilitate males and females to encounter each other in order to mate and successfully propagate the species.

## Figures and Tables

**Figure 1 insects-15-00043-f001:**
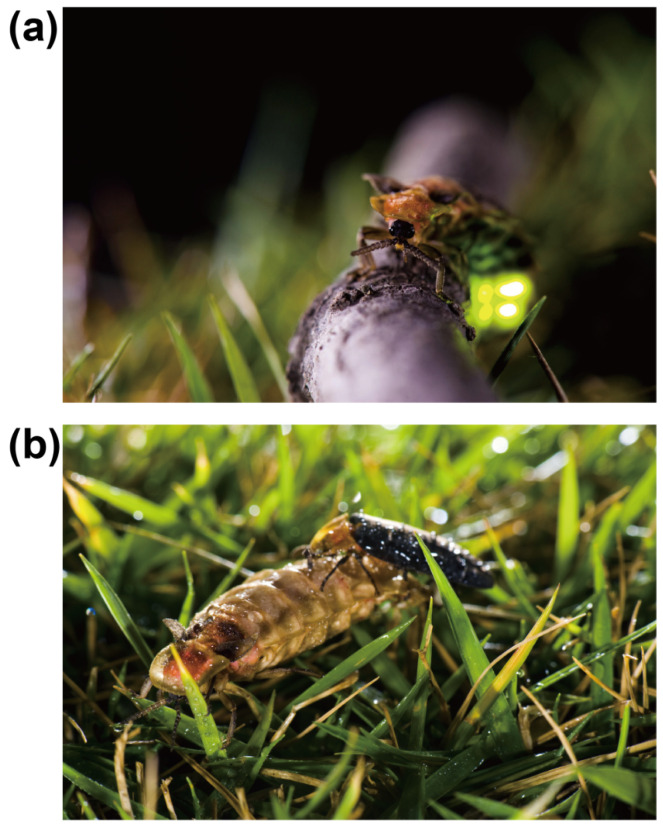
The courtship behavior of the firefly *P. pectoralis.* (**a**) A female is courting with her abdomen curling up and emitting a bright green light. (**b**) Mating, male on top of the female.

**Figure 2 insects-15-00043-f002:**
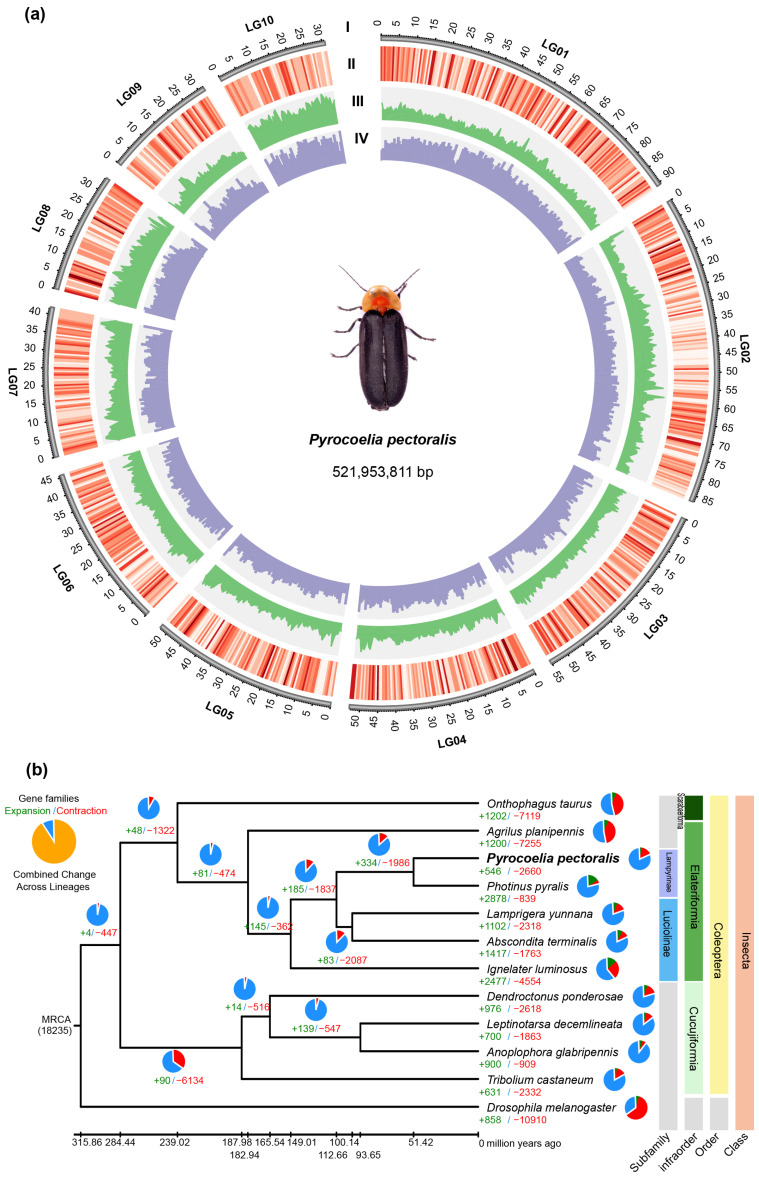
Chromosomal characteristics and phylogeny of *P. pectoralis*. (**a**) Chromosomal features. From outer to inner circles: I chromosomal, II gene density, III GC content, IV TE density, drawn in 0.5 Mb non-overlapping windows; (**b**) a maximum-likelihood phylogenetic tree is shown for *P. pectoralis* and 11 other insects. *Drosophila melanogaster* was used as the outgroup. The bootstrap value of all nodes is supported at 100/100. Support at nodes are divergence times (million years). Pie charts and numbers below represent the proportion and specific values of the gene families of expansion (green) and contraction (red), respectively.

**Figure 3 insects-15-00043-f003:**
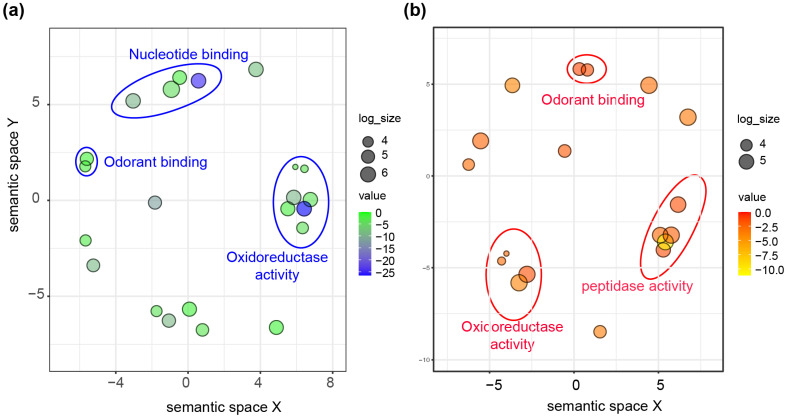
GO enrichment analysis of the expanded and species-specific orthologous family in *P. pectoralis* by Revigo. (**a**) GO terms of *P. pectoralis*-expanded genes are summarized and visualized as a scatter plot. (**b**) GO terms of *P. pectoralis* species-specific genes.

## Data Availability

The datasets generated and/or analyzed during the current study are available in this published article and the Appendix A files, the complete Hi-C genomes for *P. pectoralis* are available on NCBI GenBank with accessions JAVRBK000000000 under BioProject PRJNA1014999, and the complete mitogenome of *P. pectoralis* are available on NCBI GenBank with accessions OR790441.

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
