# Peer review of "An Improved Chromosome-Level Genome Assembly of the Firefly Pyrocoelia pectoralis"

_insects, 2024, doi:10.3390/insects15010043_

Round 1
Reviewer 1 Report
Comments and Suggestions for Authors
In this MS authors describe a new chromosome level assembly of one firefly. I have some comments to authors that I would like to be addressed:
Major:
1) When you are studying a Karyotype you should provide the chromosomes ordered by size and then, identify the sex chromosomes. In the figure you provide as "karyotype analysis...(Fig S1)" this is only a mitotic plate. Could you provide a karyotype (maybe even in the text, not as supplement) and also some images of meiosis? This is very important for further bioinformatic analysis (anchoring scaffolds to chromosomes, etc.). Can you also predict which LG is each chromosome? Were you able to assembly the Y chromosome?
2) Regarding Repeats I have several doubts regarding satDNAs. Working with assemblies instead of raw data there is a tend into a great underestimation of this kind of sequences. What I would recommend is to use Illumina data to run (at least) a run of TAREAN (or RepeatExplorer2) in order to get a better estimation of tandem repeats (and then combine this library with the rest of libraries you got). There are many studies where they prove that the estimation using assemblies always tend to underestimate this fraction in the genome.
3) A new version of the program BUSCO has been launched (v5), I would go for this one instead v3 (not mandatory but recommended). Also please provide the correct odb you are using (line 139 you say odb_9 and then in line 223 you say odb_10). (same comment for RepeatModeler, there is a new version v2)
4) Did you quality check your reads? How?
5) Regarding the previous assembly
(10.1093/gigascience/gix112)
and comparing it with the new one there is a thing that can be discussed and it is the BUSCO values (98.7% vs 93.64%). It could be good to include the total number of genes and not percentages and try to use the same odb for comparisons
Minor:
1) Line 58 a space is missing in (b)P. pectoralis
2) Line 117 from (the clean...) to line 121. Please delete this as it is again explained below.
3) Line 141 "TheClean" by "The clean"
4) Line 154 link to RepeatModeler is broken
5) Line 160 "Repeat Masker" for "RepeatMasker"
6) Line 217 "2n=18+xy" for "2n=18+XY"
7) numbers in English only have the "," when they have 5 digits or more, please correct this.
Author Response
We thank the comments to our ms and reply them in attached file"response to reviewer 1"

Reviewer 2 Report
Comments and Suggestions for Authors
The paper by Fu et al. presents improved genome assembly of the firefly Pyrocoelia pectoralis. They used Hi-C data to assemble the genome to chromosome-level. I would first like to congratulate the authors for very nicely written manuscript and high quality of presented results. They validated the firefly genome by using several tools and also expand their analysis to genetic and non-coding part. Also, the authors performed gene phylogenetic analysis on great scale, comparing with 11 other insects illuminating the presence of novel genes potentially involved with unique behavior of the fireflies. There are some minor comments bellow I would kindly ask the authors to address.
The authors noticed high redundancy and contamination of previous genome assembly that is mentioned on several places. Please add some more description somewhere on how did you detect this flaws of that assembly and to what extent have they been observed.
Line 23. Pectoralis should be in small capital letter.
Line 23. “…for the first time by our team.” - Add a reference here to your previously published genome or somehow state it refers to previously done work.
Line 58. (b)P. pectoralis - missing space
Line 62-65. This sentence seems a bit out of place and I would suggest to only mention cultural aspect of bioluminescence observation.
Line 74. “…by making us of suitable…” - Change to “…by generating suitable…”.
S1 Table could benefit from more explanation; such as what does filtered assembly genome refers to?
Line 90. Write which two species are you referring to.
Line 141. “TheClean reads...” - Is this maybe a misspelling?
Line 191. “eachortholog” - Add space.
Line 235. “…are a typical..” - “…is a typical…”
Line 252-253. 42.78% is the total repeat content from the Table S4 and not only interspersed if I understood right.
Line 256. “…accounting for 49.15% (~96.92 Mb).” - This applies to only repetitive content and not whole genome, or?
Line 267 and line 268. Maybe round up numbers for average gene/intron length.
Line 270. 20, 646 - Has extra space.
Line 332. “Compared to other firefly genomes…” - Add a bit more description of what other genomes are you referring to with appropriate references.
Author Response
The paper by Fu et al. presents improved genome assembly of the firefly Pyrocoelia pectoralis. They used Hi-C data to assemble the genome to chromosome-level. I would first like to congratulate the authors for very nicely written manuscript and high quality of presented results. They validated the firefly genome by using several tools and also expand their analysis to genetic and non-coding part. Also, the authors performed gene phylogenetic analysis on great scale, comparing with 11 other insects illuminating the presence of novel genes potentially involved with unique behavior of the fireflies. There are some minor comments bellow I would kindly ask the authors to address.
The authors noticed high redundancy and contamination of previous genome assembly that is mentioned on several places. Please add some more description somewhere on how did you detect this flaws of that assembly and to what extent have they been observed.
Answer: Line 72-77 “However, high redundancy and contamination were detected and limited its usefulness.” changed to “We detected non-firefly genes existed such as mitochondrial, bacterial, virusal, fungal and other no hit genes (Table S2) when we performed annotation of the original genome Ppec-1.0. Besides, BUSCO assessment showed the completeness of Complete and single-copy in the original genome, Ppec-1.0 is low with 60.8%, while Complete and duplicated BUSCOs (D) is high with 38.3% (Table S3). Thus far, contamination and high redundancy limited its usefulness.”
Line 23. Pectoralis should be in small capital letter.
Answer: We revised it.
Line 23. “…for the first time by our team.” - Add a reference here to your previously published genome or somehow state it refers to previously done work.
Answer: The sentence was rewritten as “A previous attempt by this team to assemble a draft genome of P. pectoralis using PacBio and Illumina HiSeq XTen platforms was limited in its usefulness by high redundancy and contamination.”
Line 58. (b)P. pectoralis - missing space
Answer: We revised it.
Line 62-65. This sentence seems a bit out of place and I would suggest to only mention cultural aspect of bioluminescence observation.
Answer: The Sentence “P. pectoralis larvae attack and kill their snail prey and are one of the snails’ major predators in Hubei Province. This is one reason why it would be important to safeguard and protect this firefly species.” was deleted.
Line 74. “…by making us of suitable…” - Change to “…by generating suitable…”.
Answer: We revised it.
S1 Table could benefit from more explanation; such as what does filtered assembly genome refers to?
Answer: Table S1 note Changed to “* The Original de novo genome is the Primary assembly genome by de novo assembly of PacBio long reads
** The original genome Ppec-1.0 is the filtered assembly genome by PacBio and Illumina HiSeq X Ten platforms selectively remove alternative heterozygous contigs
*** The Hi-C genome Ppec-2.0 is theimprovegenome assembly by Hi-C technology”
Line 90. Write which two species are you referring to.
Answer: Line 87-88 Changed to “Males and females of P. pectoralis were lab-reared for two generations at the Huazhong Agricultural University in Wuhan (China).”
Line 141. “TheClean reads...” - Is this maybe a misspelling?
Answer: We revised it.
Line 191. “eachortholog” - Add space.
Answer: We revised it.
Line 235. “…are a typical..” - “…is a typical…”
Answer: We revised it.
Line 252-253. 42.78% is the total repeat content from the Table S4 and not only interspersed if I understood right.
Answer: Line 254-255 changed to “Repeated sequences were mined and annotated. In P. pectoralis the total length of the repeats is 227.69 Mb (42.78% of the genome) (Table S6).”
Line 256. “…accounting for 49.15% (~96.92 Mb).” - This applies to only repetitive content and not whole genome, or?
Answer: Line 260-261 changed to “Among these TEs, the dominant repetitive sequence type involves the DNA transposon, accounting for 18.21% (~96.92 Mb)”
Line 267 and line 268. Maybe round up numbers for average gene/intron length.
Answer: We revised them in main text and Table S8
Line 270. 20, 646 - Has extra space.
Answer: We revised it.
Line 332. “Compared to other firefly genomes…” - Add a bit more description of what other genomes are you referring to with appropriate references.
Answer: Line 346-349 changed to “Compared with other firefly genomes, our genome assembly exhibited the least total scaffold number (363) and the highest scaffold N50 (52.87 Mb), while other firefly genome total scaffold numbers ranged from 2160 to 5365, and scaffold N50 in other firefly genomes varied from 0.69 Mb to 47.02 Mb [43].”
Round 2
Reviewer 1 Report
Comments and Suggestions for Authors
I would just change the sentence: "Twenty pairs of chromosomes were observed (2n=20)" if you see 20 pairs, it would mean a total of 40 chromosomes, so what it should be written is "Ten pair of chromosomes" or "twenty chromosomes"
Author Response
Comments and Suggestions for Authors
I would just change the sentence: "Twenty pairs of chromosomes were observed (2n=20)" if you see 20 pairs, it would mean a total of 40 chromosomes, so what it should be written is "Ten pair of chromosomes" or "twenty chromosomes"
Answer: Line 215-216, Changed to "Ten pairs of chromosomes were observed (2n=20) ......"